# Associations between physical activity types and multi-domain cognitive decline in older adults from the Three-city cohort

**Caroline Dupré**[1,2], **Catherine Helmer**[3], **Bienvenu Bongue**[1,2], **Jean François Dartigues**[3], **Frédéric Roche**[2], **Claudine Berr**[4], **Isabelle Carrière**[4]*

**1** CETAF, Saint-Etienne, France, **2** Laboratoire SNA-EPIS, University Jean Monnet, Saint-Etienne, France, **3** Inserm, Bordeaux Population Health Research Center, Univ. Bordeaux, Bordeaux, France, **4** Institute for Neurosciences of Montpellier, Inserm, Univ. Montpellier, Montpellier, France

\* isabelle.carriere@inserm.fr

**Data Availability Statement:** In the interest of participant confidentiality and to comply with the data sharing guidelines imposed by the French national data protection agency (Commission

## Abstract

Several studies suggest that physical activity improves cognitive functions and reduces cognitive decline, whereas others did not find any evidence of a neuroprotective effect. Furthermore, few cohort studies have analyzed the different physical activity types and particularly household activities. Our objective was to assess the association of two physical activity types with the decline in different cognitive domains in a large prospective cohort of community-dwelling older adults from the Three-city study. Physical activity (domestic/transportation activities and leisure/sport activities) was assessed with the Voorrips questionnaire, specific for older adults. Baseline sociodemographic and health history variables as well as cognitive performance data at baseline and during the 8-year follow-up (Mini-Mental State Examination, Benton Visual Retention Test, Trail Making Tests A and B, Isaac's Set Test and Free and Cued Selective Reminding Test) were also available. Associations between physical activity scores and cognitive decline in different domains were tested using minimally- and multi-adjusted linear mixed models. The analysis included 1697 participants without dementia at baseline and with at least one follow-up visit. At baseline, participants with higher sub-scores for the two physical activity types had better cognitive performances. Interaction with time showed that decline in some cognitive scores (Trail Making Test B and Isaac's Set Test) was significantly less pronounced in participants with higher household/transportation activity sub-scores. No significant effect over time was found for leisure/sport activities. This study shows that during an 8-year follow-up, executive functions and verbal fluency were better preserved in older adults who performed household/transportation activities at moderate to high level. Participation in domestic activities and using adapted transport means could allow older adults to maintain specific cognitive abilities.

## Introduction

During the natural process of ageing, some cognitive abilities, such as vocabulary and implicit memory, do not seem to decline and may even improve. Conversely, other abilities, such as

Nationale de l'Informatique et des Libertés, CNIL),
data are available upon request to the 3C data
access committee. Please email requests to: e3c.
coordinatingcenter@gmail.com.

**Funding:** The Three-city study is carried out under
a partnership agreement between the Institut
National de la Santé et de la Recherche Médicale
(INSERM), Victor-Segalen Bordeaux II University,
and Sanofi-Aventis. The Three-city study was also
supported by the Caisse Nationale d'Assurance
Maladie des Travailleurs Salariés, Direction
Générale de la Santé, MGEN, the Institut de la
Longévité, Agence Française de Sécurité Sanitaire
des Produits de Santé, the Regional Governments
of Aquitaine, Bourgogne and Languedoc-
Roussillon, the Fondation de France, the Ministry of
Research-Inserm Programme 'Cohorts and
collection of biological material', Novartis, the
Fondation Plan Alzheimer, the Agence Nationale de
la Recherche ANR PNR 2006 (ANR/DEDD/PNRA/
PROJ/200206-01-01), Longvie 2007 (LVIE-003-
01), the Caisse Nationale de Solidarité pour
l'Autonomie (CNSA), and Roche Pharma. This
work was supported by the Chaire Santé des aînés
et Ingénierie de la prévention, Université Jean
Monnet, Saint-Etienne, France. The funders had no
role in the study design, data collection and
analysis, or preparation of the manuscript.

**Competing interests:** The authors have declared
that no competing interests exist.

attention, executive functions, processing speed and working memory, progressively deterio-rate over time [1]. This long process is accelerated by the presence of neurodegenerative dis-eases. Furthermore, memory deficits are a central symptom in Alzheimer's disease, but semantic and verbal abilities also are impaired [2]. This leads to poor quality of life due to reduced daily life functioning and increased disability. Fewer years of education, the presence of the APOE E4 allele, traumatic brain injuries, obesity, diabetes, metabolic syndrome, hyper-tension and unhealthy diet are considered risk factors of cognitive decline [3, 4].

The protective effect of physical activity also has been investigated. In its report on reducing the risks of cognitive decline and dementia [5], the World Health Organization (WHO) con-cluded, based on low to medium quality evidence, that physical activity (PA) has a modest pro-tective effect on cognition and that this effect might be due to aerobic exercise. Baumgart et al [4] summarized the results of the independent evaluation by the Alzheimer's Association of systematic reviews and meta-analysis, longitudinal and cross-sectional studies and randomized controlled trials (RCTs) on risk factors for cognitive decline, and found that PA—even as light as walking—is associated with a reduced risk of cognitive impairment and/or improved cogni-tive functions. However, they did not determine the optimal PA duration, type and intensity. Conversely, a Cochrane review [6] found no evidence in the available data from RCTs that aer-obic PA brings cognitive benefit to cognitively healthy older adults. Although several studies suggest that PA could improve cognitive function and reduce cognitive decline [7], others found no evidence of a neuroprotective effect [8]. Moreover, few cohort studies evaluated the different PA types, particularly household activities [9]. Therefore, the debate is open on the magnitude of PA effects, the cognitive domains that might most benefit, and the most suitable PA types.

In this study, our objective was to analyze the association between two PA types and the decline over time of different cognitive domains in a large prospective cohort of older adults.

## Materials and methods

### Study design

The Three-city (3C) study is a multi-site community-living cohort of 9,294 participants aged 65 years and over. Participants were recruited between 1999 and 2001 from the electoral rolls of three French cities: Bordeaux, Dijon, and Montpellier. The aim was to study the impact of cardiovascular factors on the risk of dementia [10]. A face-to-face interview and a clinical examination were performed at inclusion and at every follow-up visit, at 2 (wave 1, W1), 4 (W2), 7 (W3), 10 (W4), 12 (W5), 15 (W6), and 17 (W7) years after inclusion (S1 Table).

The PA questionnaire was introduced at W3 for the Montpellier center and at W4 for the Bordeaux center (2515 participants). Therefore, the analyzed sample included only partici-pants who completed this questionnaire, and the baseline evaluation refers to W3 and W4 for Montpellier and Bordeaux, respectively. The study protocol was approved by the University Hospital of Kremlin-Bicêtre Ethics Committee. Each participant signed an informed consent form.

### Cognition assessment

Five cognitive tests were administered by neuropsychologists at each visit to evaluate different cognitive domains: global cognitive function was assessed with the French version of the Mini-Mental State Examination (MMSE) [11], visual working memory by the Benton Visual Reten-tion Test (BVRT) [12], psychomotor speed and executive functions by the Trail Making Test Part A and B (TMTA and TMTB) [13], verbal fluency by the Isaacs's Set Test (IST total score [14, 15] that corresponds to the sum of the number of words generated in four semantic

categories—animals, colors, cities, fruits—in 30 seconds), and verbal episodic memory by the Free and Cued Selective Reminding Test (FCSRT) [16]. For this analysis, the FCSRT "free recall score" (total number of words retrieved at the three free recall trials) and "total recall score" (total number of words retrieved at the three free and cued recall trials) were used; both scores range from 0 to 48 [17].

## Physical activity assessment

PA was assessed with the self-report Voorrips questionnaire [18] that is designed to estimate PA in older adults. This questionnaire includes three parts: household/transportation activities, leisure time activities, and sport activities. The household/transportation activity part consists of ten questions (four to five possible scores for each item) about housework, preparing meals, shopping, and transportation (car, public transportation, bicycle, walking). The total sum (divided by 10) constitutes the first sub-score used in the present analysis. The leisure time and sport activity parts include questions on the type of activity, number of hours per week, and number of months per year. The activity types are associated with intensities that are determined according to the activity energetic costs. Sitting unloaded activities, which are mainly cognitive, were excluded from the scoring in the present study. Therefore, only standing and sport activities were considered. All leisure time and sport activities were pooled in the leisure/sport activity sub-score (intensity* number of hours per week* number of months per year). As the appropriate activity level thresholds for the household/transportation and leisure/sport sub-scores were previously determined in the 3C cohort [19], participants could be classified in three groups according to their sub-scores: <1.6, [1.6;2] and >2 for household/transportation activities, and 0,] 0;8.18] and >8.18 for leisure/sport activities.

## Dementia diagnosis

At baseline and at each follow-up visit, all participants recruited in Montpellier were routinely examined by a neurologist at each follow-up visits. In Bordeaux, after an extensive cognitive and functional evaluation by a neuropsychologist specifically trained in dementia diagnosis, only participants with suspicion of dementia were examined by a neurologist. For both centers, a panel of independent neurologists expert in dementia reviewed all the existing information on the participants with suspected dementia at each visit, and a consensus on the diagnosis was obtained according to the Diagnostic and Statistical Manual of Mental Disorders, 4th edition (DSM-IV), revised criteria and etiology [20]. Participants with prevalent dementia at baseline (W3 Montpellier and W4 Bordeaux) were excluded from the present analysis.

## Baseline socio-demographic and clinical variables

Socio-demographic variables included sex, age, study center, and education level (<6 years, 6–11 years, and >11 years). Lifestyle variables were: consumption of alcohol (0.1–36 g/day, >36 g/day) and of fruits and vegetables (less than twice per day). Health status variables included diabetes (declared, treated, or glycaemia ≥7 mmol/L), body mass index (BMI, Kg/m$^2$ in three classes), self-reported cardiovascular diseases (stroke, angina pectoris, myocardial infarction, cardiac and vascular surgery), hypertension (treated, or blood pressure ≥160/95 mmHg) and depression [Center for Epidemiologic Studies Depression Scale (CES-D [21])] scores ≥16, or antidepressant treatment: Anatomical Therapeutic Chemical code N06A]. Participants with at least one ε4 allele were defined as APOE e4 carriers. Benzodiazepine use and a hierarchical disability indicator were also included. The disability indicator was calculated by combining the Rosow and Breslau Mobility Scale, Lawton-Brody Instrumental Activity of Daily Living (IADL) scale, and Katz Activity of Daily Living (ADL) scale [22].

## Statistical methods

Baseline characteristics were compared between included and excluded participants using the chi-square and Wilcoxon rank-sum tests.

To keep in the analysis participants with missing baseline data, multiple imputations were carried out using a fully conditional specification method with discriminant function for the nominal variable [23]. The missing values represented 2% of all data for 28 variables. The SAS PROC MI procedure [24] with the FCS DISCRIM option for categorical variables and five imputations was used.

The participants' characteristics were compared according to their household/transportation and leisure/sport activity sub-score group, and differences were tested with the Chi-square and Wilcoxon rank-sum tests. Associations between baseline PA groups and changes in cognitive performances were analyzed using linear mixed regression models. Separate models were performed for each cognitive test with each cognitive score as dependent variable (baseline and follow-up measures). As several cognitive scores had a skewed distribution, they were transformed using the most adapted functions: $(15\text{-BRVT})^{1/2}$, $(30\text{-MMSE})^{1/2}$, natural logarithm of the TMT scores, and $(48\text{-FCSRT total recall score})^{1/2}$. The effect of each potential confounding variable (main-effect term and interaction-with-time term) was also examined separately with a linear mixed model adjusted for sex, age, center, and education level. Covariates were selected if significant ($p < 25\%$) for at least one term and one cognitive test. First, the relation with the three PA sub-score groups was adjusted for age at baseline, sex, study center and education level, and their interaction with time for age and study center (minimally adjusted model). Then, the model was additionally adjusted for the selected covariates (consumption of fruits and vegetables, alcohol, BMI, hypertension, diabetes, cardiovascular disease, APOE carrier, depression and benzodiazepine use) and their time interactions if significant (for BMI, hypertension, diabetes, APOE carrier, depression). Interactions between PA groups and sex, age and educational level were also tested. The mixed model analyses were carried out using the five imputed datasets. The estimates were then combined using the SAS MIANALYZE procedure. Additional sensitivity analyses were performed to address possible reverse causality. Specifically, models were restricted to participants without IADL/ADL limitations at baseline and also to those free of dementia at all follow-up visits. All statistical analyses were carried out with SAS, version 9.4.

## Results

Among the 2515 participants at baseline, 244 were not included because of dementia at baseline and 124 because they were confined at home. In addition, 29 participants were excluded because of missing baseline cognitive tests, 276 because they did not have any follow-up visit, and 145 due to absence of the two baseline PA sub-scores. Finally, 1697 participants were included in the analysis. Excluded individuals were older ($p < 0.0001$), less likely to be women ($p = 0.0126$), with lower education level ($p = 0.0034$), less fully independent ($p < 0.0001$), and with more depressive symptoms ($p = 0.0002$).

### Sample description

At baseline, the median age was 79.7 years (IQR 76.9–83.0), 63.5% of participants were women, and 17.5% had at least one APOE e4 allele. Moreover, 10.1% consumed fruits and vegetables less than twice per day and 10.6% were obese. Concerning comorbidities, 68.7% had blood hypertension, 10.4% diabetes, 16.3% had cardiovascular diseases, 17.9% had depression, and 18% took benzodiazepines. When participants were divided in three groups according to their PA sub-scores, being older, dependent or obese or having hypertension was significantly

associated with lower PA scores both for household/transportation and leisure/sport activities. Diabetes and cardiovascular diseases were only associated with lower household/transportation PA scores, while low educational level, low fruit and vegetable consumption, depression, and taking benzodiazepines were only associated with lower leisure/sport PA scores. Regarding sex distribution, the percentage of women was significantly higher in the group with the highest household/transportation PA sub-score and lower in the group with the highest leisure/sport PA sub-score. Alcohol consumption was associated with lower household/transportation PA sub-score and higher leisure/sport PA sub-score (Table 1 and S2 Table). All baseline cognitive scores were significantly better in participants with higher PA sub-scores, except for the BVRT and FCSRT total recall scores that did not differ in the three household/transportation (p = 0.06) and leisure/sport activity PA groups (p = 0.76), respectively.

### Physical activity and cognitive decline over the 8 years of follow-up

The median (IQR) follow-up time was 7.9 years (7.5–8.1). The results of the model adjusted for sex, age, center and education level (Table 2) showed that at baseline, participants in the intermediate household/transportation PA group had better BRVT, TMTA, TMTB, IST and FCSRT-free recall scores (but TMTA and FCSRT-free recall scores were better only in the highest sub-score group). Similarly, intermediate and higher leisure/sport PA sub-scores were associated with better TMTA, TMTB, IST and FCSRT-free recall scores. The interaction with time showed that the cognitive performance assessed by the TMTB and IST decreased slower over time in participants in the highest household/transportation PA group and for TMTB also in the intermediate group. Conversely, no significant effect over time was found for the leisure/sport activities.

The second model (Table 3), adjusted for all selected covariates and their interaction with time, if significant, showed that the PA sub-scores were no longer significantly associated with the baseline BRVT and TMTB scores. The interaction between household/transportation PA groups and time remained significant for the TMTB (p = 0.03) and IST (p = 0.009) scores, suggesting a slower decline over time, and was borderline significant for the TMTA scores (p = 0.06). For instance, according to the parameters estimated by this multi-adjusted model, in 80-year-old participants from Bordeaux without comorbidity (obesity, hypertension, diabetes, depression) and without ApoE allele 4, the mean IST decline was 0.90 word per year in the low-level and 0.69 word per year in the high-level household/transportation PA group. For TMTB, in the same participants, the mean time in seconds to perform the test was multiplied by 1.049 and by 1.039 each year in the low-level and high-level household/transportation PA groups, respectively. For a baseline TMTB score of 100 seconds, this would correspond approximatively to an increase of 4.9 and 3.9 seconds per year, respectively. Again, no significant effect over time was found for the leisure/sport activities.

The interactions of sex, age, and education with the household/transportation and leisure/sport PA groups were not significant, including the interaction with time.

### Sensitivity analyses

When only participants without IADL/ADL limitations at baseline were included (n = 1461), the interaction between household/transportation PA groups with time remained significant for the TMTB and IST scores in the multi-adjusted models. Moreover, the interaction between household/ transportation PA scores and time became significant also for the TMTA scores (β (SE) = -0.01(0.004) and p = 0.014)). Whatever the cognitive function analyzed, no significant interaction with time was observed for the leisure/sport PA groups.

**Table 1. Participants' characteristics according to their level of household and transportation-related physical activities, n = 1697.**

| | Baseline household and transportation activities | | | Chi² test p value |
|---|---|---|---|---|
| | ≤1.6 | ] 1.6–2.0] | >2.0 | |
| | n = 622 | n = 580 | n = 495 | |
| | % | % | % | |
| Sex, female | 44.53 | 66.72 | 83.64 | < .0001 |
| Education level | | | | 0.3088 |
| <6 years | 23.67 | 22.11 | 22.06 | |
| 6–11 years | 25.44 | 30.74 | 29.96 | |
| >11 years | 50.89 | 47.15 | 47.98 | |
| Hierarchical disability indicator | | | | < .0001 |
| Fully independent | 34.38 | 42.04 | 51.4 | |
| Mild disability | 41.01 | 46.48 | 44.3 | |
| Moderate to severe disability | 24.61 | 11.48 | 4.3 | |
| Fruit and vegetable consumption | | | | 0.71 |
| less than twice per day | 10.78 | 10.16 | 9.26 | |
| Alcohol | | | | 0.0095 |
| 0 | 29.45 | 31.09 | 36.84 | |
| 1–36 g/day | 65.21 | 65.8 | 60.73 | |
| > 36 g/day | 5.34 | 3.11 | 2.43 | |
| Body mass index | | | | < .0001 |
| Normal (<25) | 43.79 | 56.24 | 63.43 | |
| Overweight (25–30) | 42.16 | 33.74 | 29.75 | |
| Obese (≥30) | 14.05 | 10.02 | 6.82 | |
| Treated hypertension or blood pressure ≥160/95 mm Hg | 76.05 | 67.41 | 60.81 | < .0001 |
| Diabetes | 14.31 | 10.55 | 5.3 | < .0001 |
| Cardiovascular disease | 22.03 | 15.69 | 9.7 | < .0001 |
| Depressive symptoms (CES-D ≥16 or treatment) | 16.67 | 17.77 | 19.46 | 0.4893 |
| Benzodiazepine use | 20.26 | 16.55 | 16.77 | 0.1752 |
| APOE4 allele | 14.14 | 16.82 | 22.27 | 0.0021 |
| | Median (IQR) | Median (IQR) | Median (IQR) | Wilcoxon test |
| Age | 81 (74–87) | 80 (73–86) | 78 (72–84) | < .0001 |
| MMSE score | 28 (26–30) | 28 (26–30) | 29 (27–31) | 0.03 |
| BVRT score | 12 (9–15) | 12 (10–14) | 12 (10–14) | 0.06 |
| TMTA (in seconds) | 51 (24–78) | 47 (23–71) | 46 (23–69) | 0.0002 |
| TMTB (in seconds) | 103 (36–170) | 94 (40–148) | 98 (46–150) | 0.01 |
| IST score | 45 (30–60) | 48 (34–62) | 48 (35–61) | 0.0002 |
| FCSRT "free recall score" | 25 (15–35) | 26 (18–34) | 26 (18–34) | < .0001 |
| FCSRT "total recall score" | 46 (41–51) | 47 (43–51) | 47 (44–50) | 0.0003 |

BVRT: Benton Visual Retention Test, FCSRT: Free and Cued Selective Reminding Test, IQR: interquartile range, IST: Isaacs Set Test, MMSE: Mini-Mental State Examination, TMTA or TMTB: Trail Making Tests A or B.

When the incident cases of dementia during the follow-up were excluded, the results for the interaction with time were similar for the TMTB [$\beta$(SE) = -0.01(0.008), p = 0.054] and IST scores [$\beta$(SE) = 0.15(0.08), p = 0.061].

## Discussion

In this prospective cohort of community-dwelling participants older than 72 years of age, the decline over time in the TMTB (executive functions) and IST (verbal fluency) was significantly

**Table 2. Association of physical activity with cognitive changes (minimally adjusted model\*).**

| | MMSE | | BVRT | | TMTA | | TMTB | | IST | | FCSRT free recall score | | FCSRT total recall score | |
|---|---|---|---|---|---|---|---|---|---|---|---|---|---|---|
| | sqrt(30-MMSE) | | sqrt(15-BVRT) | | ln(TMTA) | | ln(TMTB) | | | | | | sqrt(48- FCSRT) | |
| | N = 1691 | | N = 1621 | | N = 1569 | | N = 1527 | | N = 1657 | | N = 1594 | | N = 1593 | |
| | beta (SE) | P value | beta (SE) | P value | beta (SE) | P value | beta (SE) | P value | beta (SE) | P value | beta (SE) | P value | beta (SE) | P value |
| Household/transportation activities | | | | | | | | | | | | | | |
| ] 1.6–2.0] | -0.009* (0.039) | 0.823 | -0.063 (0.031) | 0.039 | -0.051 (0.020) | 0.011 | -0.047 (0.022) | 0.037 | 1.596 (0.542) | 0.003 | 1.092 (0.366) | 0.003 | -0.079 (0.064) | 0.214 |
| >2.0 | -0.064 (0.043) | 0.138 | -0.039 (0.033) | 0.235 | -0.051 (0.022) | 0.021 | -0.011 (0.024) | 0.649 | 1.053 (0.590) | 0.074 | 0.861 (0.396) | 0.030 | -0.061 (0.069) | 0.375 |
| Household/transportation activities * time | | | | | | | | | | | | | | |
| ] 1.6–2.0] | 0.002 (0.008) | 0.831 | 0.005 (0.006) | 0.430 | -0.005 (0.004) | 0.142 | -0.009 (0.004) | 0.031 | 0.064 (0.076) | 0.401 | 0.041 (0.071) | 0.568 | -0.007 (0.013) | 0.610 |
| >2.0 | 0.011 (0.009) | 0.210 | 0.004 (0.007) | 0.539 | -0.007 (0.004) | 0.067 | -0.010 (0.004) | 0.018 | 0.168 (0.079) | 0.033 | 0.025 (0.072) | 0.733 | -0.004 (0.013) | 0.736 |
| Leisure/sports activities | | | | | | | | | | | | | | |
| ] 0–8.18] | -0.054 (0.037) | 0.146 | -0.006 (0.029) | 0.840 | -0.065 (0.019) | 0.001 | -0.043 (0.022) | 0.047 | 1.885 (0.514) | 0.0002 | 1.092 (0.344) | 0.002 | -0.072 (0.061) | 0.233 |
| >8.18 | -0.048 (0.047) | 0.315 | -0.001 (0.037) | 0.976 | -0.012 (0.024) | < .0001 | -0.053 (0.027) | 0.047 | 2.400 (0.658) | 0.0003 | 1.133 (0.438) | 0.010 | -0.017 (0.077) | 0.828 |
| Leisure/sports activities * time | | | | | | | | | | | | | | |
| ] 0–8.18] | -0.003 (0.008) | 0.671 | -0.001 (0.006) | 0.918 | -0.002 (0.004) | 0.523 | -0.005 (0.004) | 0.204 | -0.078 (0.074) | 0.290 | -0.104 (0.067) | 0.120 | 0.019 (0.012) | 0.128 |
| >8.18 | -0.004 (0.010) | 0.662 | -0.006 (0.007) | 0.438 | -0.001 (0.004) | 0.732 | -0.008 (0.005) | 0.089 | -0.074 (0.091) | 0.415 | -0.038 (0.083) | 0.647 | -0.001 (0.015) | 0.951 |

\**Models adjusted for age, sex, study center, education level, and time by age and time by study center interactions.*

SE standard error, sqrt: square root, ln: natural logarithm.

BVRT: Benton Visual Retention Test, FCSRT: Free and Cued Selective Reminding Test, IST: Isaacs Set Test, MMSE: Mini-Mental State Examination, TMTA or TMTB: Trail Making Tests A or B.

slower in participants who reported moderate or high level of household/transportation-related PA, whereas the TMTA (psychomotor speed) performance change was almost significant. At baseline and in the multi-adjusted models, the two PA sub-scores (household/transportation and leisure/sports activities) were also positively associated with better TMTA, IST and FCSRT ("free recall") scores. During the follow-up, the increase in the time required to perform the TMTB was almost one second shorter per year, while the decline in the number of generated words in the IST was reduced by approximately 0.2 words per year in the high-level household/transportation activity group compared with the low-level household/transportation activity group. These longitudinal results remained significant after adjustment for a large number of variables, including diabetes, cardiovascular diseases and hypertension, and also in the sensitivity analyses that excluded participants with incident dementia or moderate to severe disability.

Our study has several limitations. First, in our aged sub-sample of the 3C cohort (median = 80 years), the level of leisure and sports activities was relatively low. Although this was representative of PA in this age category, 40.4% of them did not perform any activity of this type, and this may preclude the detection of a significant PA effect. Second, their low PA level could be also a consequence of comorbidities or frailty. However, the effect of household/transportation-related PA persisted after adjustment for many different comorbidities and risk

**Table 3. Associations of physical activity with cognitive changes (multi-adjusted model*).**

| | MMSE | | BVRT | | TMTA | | TMTB | | IST | | FCSRT free recall score | | FCSRT total recall score | |
|---|---|---|---|---|---|---|---|---|---|---|---|---|---|---|
| | sqrt(30-MMSE) | | sqrt(15-BVRT) | | ln(TMTA) | | ln(TMTB) | | | | | | sqrt(48- FCSRT) | |
| | N = 1691 | | N = 1621 | | N = 1569 | | N = 1527 | | N = 1657 | | N = 1594 | | N = 1593 | |
| | beta (SE) | P value | beta (SE) | P value | beta (SE) | P value | beta (SE) | P value | beta (SE) | P value | beta (SE) | P value | beta (SE) | P value |
| Household/transportation activities | | | | | | | | | | | | | | |
| ] 1.6–2.0] | 0.009 (0.039) | 0.826 | -0.054 (0.031) | 0.078 | -0.045 (0.020) | 0.024 | -0.038 (0.022) | 0.092 | 1.293 (0.539) | 0.017 | 0.992 (0.365) | 0.007 | -0.069 (0.064) | 0.285 |
| >2.0 | -0.039 (0.044) | 0.369 | -0.026 (0.034) | 0.447 | -0.041 (0.022) | 0.063 | 0.00004 (0.025) | 0.999 | 0.632 (0.595) | 0.288 | 0.705 (0.400) | 0.078 | -0.052 (0.071) | 0.466 |
| Household/transportation activities * time | | | | | | | | | | | | | | |
| ] 1.6–2.0] | -0.001 (0.008) | 0.907 | 0.005 (0.006) | 0.463 | -0.006 (0.004) | 0.120 | -0.009 (0.004) | 0.035 | 0.095 (0.076) | 0.215 | 0.054 (0.072) | 0.450 | -0.010 (0.013) | 0.442 |
| >2.0 | 0.007 (0.009) | 0.462 | 0.003 (0.007) | 0.645 | -0.007 (0.004) | 0.060 | -0.009 (0.004) | 0.032 | 0.210 (0.081) | 0.009 | 0.054 (0.075) | 0.473 | -0.011 (0.014) | 0.429 |
| Leisure/sports activities | | | | | | | | | | | | | | |
| ] 0–8.18] | -0.027 (0.038) | 0.468 | 0.024 (0.030) | 0.427 | -0.052 (0.019) | 0.007 | -0.021 (0.022) | 0.342 | 1.383 (0.522) | 0.008 | 0.847 (0.349) | 0.015 | -0.066 (0.062) | 0.289 |
| >8.18 | -0.004 (0.048) | 0.941 | 0.042 (0.037) | 0.263 | -0.082 (0.024) | 0.001 | -0.019 (0.027) | 0.470 | 1.612 (0.669) | 0.016 | 0.792 (0.447) | 0.076 | 0.0004 (0.079) | 0.996 |
| Leisure/sports activities * time | | | | | | | | | | | | | | |
| ] 0–8.18] | -0.003 (0.008) | 0.747 | -0.003 (0.006) | 0.688 | -0.002 (0.004) | 0.563 | -0.005 (0.004) | 0.236 | -0.068 (0.075) | 0.366 | -0.111 (0.068) | 0.101 | 0.021 (0.013) | 0.086 |
| >8.18 | -0.003 (0.010) | 0.805 | -0.007 (0.008) | 0.341 | -0.001 (0.004) | 0.870 | -0.008 (0.005) | 0.116 | -0.066 (0.094) | 0.483 | -0.066 (0.086) | 0.444 | 0.005 (0.016) | 0.752 |

*Models adjusted for age, sex, study center, education, depression, alcohol, BMI, benzodiazepine, diabetes, hypertension, APOE 4 allele, consumption of fruits and vegetables (less than twice per day), cardiovascular disease, and time by age, study center, depression, BMI, diabetes, hypertension and APOE4 allele interactions.

SE standard error, sqrt: square root, ln: natural logarithm.

BVRT: Benton Visual Retention Test, FCSRT: Free and Cued Selective Reminding Test, IST: Isaacs Set Test, MMSE: Mini-Mental State Examination, TMTA or TMTB: Trail Making Tests A or B.

factors. Third, low PA could also be a consequence of a pre-dementia state. However, the long follow-up (8 years) and the sensitivity analysis after exclusion of participants with incident dementia limited this possibility. Fourth, PA was measured using a self-report questionnaire that might be susceptible to information bias (misreport of activity or frequency and duration overestimation). However, the Voorrips questionnaire has been specially designed for older adults and has been validated for light and high activities [25]. Finally, each cognitive test was analyzed separately and this could raise the issue of test multiplicity that might increase the false positive rate. However, each cognitive test assessed a specific cognitive domain, and PA might influence these cognitive domains in different ways.

Our study has several strengths. First, this was a longitudinal and multi-center study with a large sample size (n = 1697). Cognition was assessed with different tests that allowed evaluating global cognition and specific cognitive domains (working memory, executive functions, verbal fluency, verbal episodic memory). PA was assessed using the Voorrips questionnaire that is specific for older adults and allows analyzing two PA types: household/transportation activities and leisure/sports activities.

Many studies, systematic reviews, and meta-analyses have evaluated the link between PA and cognition. The systematic review by Cunningham et al. [26] found a reduction in the risk of cognitive decline (by 26% for moderate and by 33% for high PA). Conversely, the systematic review by Kikkert et al. [27] did not find PA as protector of cognitive decline. A meta-analysis of five randomized trials [28] concluded that PA was not associated with cognitive decline. These discrepancies may be due to the heterogeneity of the methods used. Some studies [29–35] used logistic regression analysis, and therefore did not consider the effect of time on the occurrence of cognitive disorders. Others included only women [32, 36] or only men [37]. In some studies [33–35, 38], the very short follow-up period did not allow reaching any conclusion. In other analyses [39–44], contradictory results were obtained due to differences in study design/statistical methods, cognitive test(s) used, or PA type analyzed. Therefore, it is necessary to develop models that take into account time and comorbidities, and with a sufficiently long follow-up, as done in our cohort. The cognitive domains evaluated and the tests used also might explain the result heterogeneity. Two indicators of global cognition may show different results. For instance, in the study by Willey et al [45] PA was significantly linked to global cognition defined by the MMSE score, but not with the Telephone Interview for Cognitive Status that is more sensitive to memory changes [46]. Noticeably, two studies [42, 47] showed a possible PA protective effect particularly on episodic memory and language.

Our study did not detect a positive effect of sports and leisure activities on cognition, as demonstrated by a recent study [8]. Conversely, it found a positive effect of domestic activities on verbal fluency and executive functions during the 8 years of follow-up, in agreement with previous findings [29, 48, 49]. Newson et al [49] showed that in older adults, lifestyle activities, such as household, domestic and social activities, influence some cognitive domains, particularly speed of processing and picture naming. Angevaren et al [48] associated PA intensity with processing speed; they defined PA by leisure activities (walking, cycling, housekeeping, gardening). It has been proposed that PA influence some cognitive functions, such as processing speed, more than memory and mental flexibility that require also knowledge and experience [49]. Household and transportation activities are very important for older adults because they represent a large part of their daily activities. These regular, non-intensive activities may have a stronger protective effect on cognition than sport activities that are far less frequent. In agreement, it has been reported that light, regular aerobic exercise increases neurogenesis and neuroplasticity and improves cardiovascular function and its associated influence on the cerebrovascular system [50].

The prevalence of older adults reporting insufficient PA varies widely around the world, between rural and urban societies, and between men and women, and is increasing in high-income countries [51]. This might lead to differences in the statistical power of the studies, and could explain some inconsistent previous findings. The type of PA practiced also is changing in Western countries, with less time spent performing domestic activities and more time dedicated to leisure activities. However, these changes are not uniform across countries due to cultural norms and social habits. For example, the proportion of people performing resistance training or strength exercises is lower in South-eastern than in Nordic European countries [52]. Therefore, the results of our study highlighting the importance of domestic activities should be confirmed in other older populations.

## Conclusion

We found a slower decline of cognitive functions, particularly executive functions and verbal fluency, over a 8-year follow-up period, in ≥72-year-old people who performed moderate to high household/transportation activities. Conversely, we did not detect any association with

leisure and sports activities. These results remained stable after adjustment for potential confounders. Our study shows the importance of considering the PA type using a specific questionnaire that includes also domestic activities. Continuing to participate in domestic activities and to use adapted transport could allow older adults to better maintain their cognitive capacities.

## Supporting information

**S1 Table. 3C cohort design.**
(PDF)

**S2 Table. Participants' characteristics according to their leisure and sport activities level, n = 1697.**
(PDF)

## Author Contributions

**Conceptualization:** Claudine Berr, Isabelle Carrière.

**Formal analysis:** Caroline Dupré.

**Methodology:** Caroline Dupré, Catherine Helmer, Claudine Berr, Isabelle Carrière.

**Supervision:** Catherine Helmer, Bienvenu Bongue, Jean François Dartigues, Frédéric Roche, Claudine Berr, Isabelle Carrière.

**Writing – original draft:** Caroline Dupré, Isabelle Carrière.

**Writing – review & editing:** Catherine Helmer, Bienvenu Bongue, Jean François Dartigues, Frédéric Roche, Claudine Berr.

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
