## [Decision Letter · Decision Letter 0]

26 Mar 2021

PONE-D-21-03103

Household- and transportation-related physical activities are prospectively associated with preserved executive functions and verbal fluency in older people from the Three-city cohort.

PLOS ONE

Dear Dr. Carriere,

Thank you for submitting your manuscript to PLOS ONE. After careful consideration, we feel that it has merit but does not fully meet PLOS ONE’s publication criteria as it currently stands. Therefore, we invite you to submit a revised version of the manuscript that addresses the points raised during the review process.

Two Reviewers well assessed this manuscript.  However, several major revisions are needed in the present form.  See the Reviewers’ comments and respond them appropriately.

We look forward to receiving your revised manuscript.

Kind regards,

Masaki Mogi

Academic Editor

PLOS ONE

Journal Requirements:

"The Three-city study is conducted under a partnership agreement between the Institut National de la Santé et de la Recherche Médicale (INSERM), Victor-Segalen Bordeaux¬ II University, and Sanofi- Aventis. The Three-city study was also supported by the Caisse Nationale d'Assurance Maladie des Travailleurs Salariés, Direction Générale de la Santé, MGEN, the Institut de la Longévité, Agence Française de Sécurité Sanitaire des Produits de Santé, the Regional Governments of Aquitaine, Bourgogne and Languedoc-Roussillon, the Fondation de France, the Ministry of Research-Inserm Programme 'Cohorts and collection of biological material', Novartis, the Fondation Plan Alzheimer, the Agence Nationale de la Recherche ANR PNR 2006 (ANR/DEDD/PNRA/PROJ/200206-01-01) and Longvie 2007 (LVIE- 003-01), the Caisse Nationale de Solidarité pour l'Autonomie (CNSA) and the Roche Pharma. This work was supported by the Chaire Santé des aînés et Ingénierie de la prévention, Université Jean Monnet, Saint-Etienne, France

The funders had no role in study design, data collection and analysis, or preparation of the manuscript."

"The authors received no specific funding for this work."

Reviewers' comments:

Reviewer's Responses to Questions

**Comments to the Author**

1. Is the manuscript technically sound, and do the data support the conclusions?

Reviewer #1: Yes

Reviewer #2: Yes

2. Has the statistical analysis been performed appropriately and rigorously? 

Reviewer #1: Yes

Reviewer #2: Yes

3. Have the authors made all data underlying the findings in their manuscript fully available?

Reviewer #1: Yes

Reviewer #2: Yes

4. Is the manuscript presented in an intelligible fashion and written in standard English?

Reviewer #1: Yes

Reviewer #2: Yes

5. Review Comments to the Author

Reviewer #1: 1. The title of the article can not cover the research content. The title is “Household- and transportation-related physical activities are prospectively associated with preserved executive functions and verbal fluency in older people from the Three-city cohort”, but this article investigates the effects of household/transportation activities, leisure time activities, and sport activities on the cognitive function of the elderly. And the cognitive domain is more than executive function and language fluency. 2. Introduction part should be further organized to make it clearer, more logical, and hierarchical.

3.The Voorrips questionnaire includes three parts: household/transportation activities, leisure time activities, and sport activities. Why do you divide the physical activity into two categories( household/transportation activities, leisure/sport activities) in your paper? What is the theoretical basis for this classification?

4. In the paper, you used the Mini-Mental State Examination (MMSE). Although the Mini-Mental State Examination (MMSE) was widely available from 1975 it is under copyright and since 2001 the authors have licensed Psychological Assessment Resources (PAR) to enforce this and payment to use this scale is required. As you used MMSE in the research reported in your submission you need to provide us with evidence of the payment/license.

5. On page 5, the word "Wave" is not clearly explained. Can you provide the raw data in the supplementary document?

6.The discussion part is too superficial, it is only a repetition of the research results, and there is no further analysis and explanation of the results.

7. The subjects of this study are about 80 years old, their level of leisure and sports activities was relatively low(research limitation 1). They can not represent the whole elderly population. Therefore, the expression of the conclusion in this paper should be adjusted.

Reviewer #2: This article is of great scientific and clinical use value.

Overall the manuscript, from the Introduction to the Conclusion, is easy to

read, the text is well written, concise and logical and, conceptually and

methodologically, relevant and correct.

Results, discussion and conclusions are very well organized and written, supported by recent and relevant scientific literature.

The manuscript, however, suffers from minor errors, that authors can improve.

I outlined some (terminology) and I suggested to make some changes (please, see comments).

In discussion, if possible, introduce the sociocultural factors to explain some differences from the other studies once it was conducted with French older adults, probably different from Northern European or anglossaxonic populations.

6. PLOS authors have the option to publish the peer review history of their article (what does this mean?). If published, this will include your full peer review and any attached files.

Reviewer #1: **Yes: **Xing WANG

Reviewer #2: No

---

## [Author Response · Author response to Decision Letter 0]

28 Apr 2021

Academic editor's comments:

The author list, affiliations and the section headings were revised. The naming of supplemental tables and files was also changed.

Our database includes information, such as age, sex, city of residence (study center), education levels, that are considered as personal data according to French and European laws. In the interest of participant confidentiality and in keeping with the data sharing guidelines imposed by the French national data protection agency (Commission Nationale de l’Informatique et des Libertés, CNIL), the data from the 3C cohort used in this study are available upon request. Interested researchers may contact e3c.coordinatingcenter@gmail.com for access.

We modified the statement as follows:

" In the interest of participant confidentiality and to comply with the data sharing guidelines imposed by the French national data protection agency (Commission Nationale de l’Informatique et des Libertés, CNIL), data are available upon request to the 3C data access committee. Please email requests to: e3c.coordinatingcenter@gmail.com"

Please remove any funding-related text from the manuscript and let us know how you would like to update your Funding Statement. 

The Funding statement was removed from the manuscript text and the online Funding statement was replaced by:

"The Three-city study is carried out under a partnership agreement between the Institut National de la Santé et de la Recherche Médicale (INSERM), Victor-Segalen Bordeaux II University, and Sanofi-Aventis. The Three-city study was also supported by the Caisse Nationale d'Assurance Maladie des Travailleurs Salariés, Direction Générale de la Santé, MGEN, the Institut de la Longévité, Agence Française de Sécurité Sanitaire des Produits de Santé, the Regional Governments of Aquitaine, Bourgogne and Languedoc-Roussillon, the Fondation de France, the Ministry of Research-Inserm Programme 'Cohorts and collection of biological material', Novartis, the Fondation Plan Alzheimer, the Agence Nationale de la Recherche ANR PNR 2006 (ANR/DEDD/PNRA/PROJ/200206-01-01), Longvie 2007 (LVIE-003-01), the Caisse Nationale de Solidarité pour l'Autonomie (CNSA), and Roche Pharma. This work was supported by the Chaire Santé des aînés et Ingénierie de la prévention, Université Jean Monnet, Saint-Etienne, France.

The funders had no role in the study design, data collection and analysis, or preparation of the manuscript."

Reviewer #1: 

1. The title of the article cannot cover the research content. The title is “Household- and transportation-related physical activities are prospectively associated with preserved executive functions and verbal fluency in older people from the Three-city cohort”, but this article investigates the effects of household/transportation activities, leisure time activities, and sport activities on the cognitive function of the elderly. And the cognitive domain is more than executive function and language fluency. 

We thank the reviewer for this comment. We changed the title as follows:

"Associations between physical activity types and multi-domain cognitive decline in older adults from the Three-city cohort."

2. Introduction part should be further organized to make it clearer, more logical, and hierarchical.

We added a transition sentence before the focus on the protective effect of physical activity and we now better define the objective. 

3.The Voorrips questionnaire includes three parts: household/transportation activities, leisure time activities, and sport activities. Why do you divide the physical activity into two categories( household/transportation activities, leisure/sport activities) in your paper? What is the theoretical basis for this classification?

The Voorrips questionnaire collects data in two ways: 

- For household and transportation activities, the questionnaire includes 10 multiple-choice questions about housework, size of the house, cooking and use of transport. Each question has 3 or 4 possible answers. This first part of the questionnaire was used to calculate the household/transportation score. 

- For leisure and sport activities, two open-ended questions ask the participants to give their activities and then the frequencies. Thus, the same activity can be cited as leisure or sport activity. For example, several participants cited cycling in the sport part and others in the leisure part of the questionnaire. 

We could not reclassify leisure and sport activities according to their intensity because the Voorrisp questionnaire does not collect this information. In addition, in French older adults (≥72 years of age), intense sport activities are unusual. Therefore, we considered that the classification as sport or leisure activities by the participants was not accurate, and we pooled the two parts of the questionnaire into a single score.

4. In the paper, you used the Mini-Mental State Examination (MMSE). Although the Mini-Mental State Examination (MMSE) was widely available from 1975 it is under copyright and since 2001 the authors have licensed Psychological Assessment Resources (PAR) to enforce this and payment to use this scale is required. As you used MMSE in the research reported in your submission you need to provide us with evidence of the payment/license.

We used the original French version of the questionnaire that is unlicensed.

5. On page 5, the word "Wave" is not clearly explained. Can you provide the raw data in the supplementary document?

To clarify the 3C cohort design, we now provide in a supplemental table, for each wave and each center, the number of participants still in the cohort and the observed time since inclusion. For practical reasons and also because the sample was aged, the examination time varies from one participant to another in the same wave. However, we used continuous-time statistical models to take into account this variation. 

6.The discussion part is too superficial, it is only a repetition of the research results, and there is no further analysis and explanation of the results.

This comment seems to contradict the analysis by reviewer #2 who wrote that "Results, discussion and conclusions are very well organized and written, supported by recent and relevant scientific literature." In the Discussion we compared our results with the abundant literature in the field and suggested possible explanations for conflicting results. The inconsistencies can be due to inadequate methods, analysis of different physical activity types, and assessment of cognitive abilities related to different domains. We extensively discussed the limits and the strengths of our study. 

We also added in the new version a paragraph at the end of the Discussion section that mentions the differences in both prevalence and type of physical activities in different world regions, between rural and urban societies, and between genders. This may lead to differences in the statistical power of the studies and explain some inconsistent previous findings. We also highlighted that the physical activity types are changing with a decrease of domestic activities and increase of leisure time, and that these changes are not uniform across countries. We added two references and concluded on the need to confirm our results in other older populations.

7. The subjects of this study are about 80 years old, their level of leisure and sports activities was relatively low (research limitation 1). They cannot represent the whole elderly population. Therefore, the expression of the conclusion in this paper should be adjusted.

We agree that our results only apply to people older than 72 years, and we clarified this point in the conclusion. 

Reviewer #2: 

1- This article is of great scientific and clinical use value. Overall the manuscript, from the Introduction to the Conclusion, is easy to read, the text is well written, concise and logical and, conceptually and methodologically, relevant and correct.

Results, discussion and conclusions are very well organized and written, supported by recent and relevant scientific literature.

We thank the reviewer for his/her very positive comments on our manuscript.

2- The manuscript, however, suffers from minor errors, that authors can improve. I outlined some (terminology) and I suggested to make some changes (please, see comments).

The errors have been corrected in the new version. 

3- In discussion, if possible, introduce the sociocultural factors to explain some differences from the other studies once it was conducted with French older adults, probably different from Northern European or anglossaxonic populations.

We thank the reviewer for this suggestion. We have written an additional paragraph at the end of the Discussion section that mentions the differences in both prevalence and type of physical activities in different world regions, between rural and urban societies, and between genders. This may lead to differences in the statistical power of the studies and explain some inconsistent previous findings. We also highlighted that the physical activity types are changing with a decrease of domestic activities and increase of leisure time, and that these changes are not uniform across countries. We added two references and concluded on the need to confirm our results in other older populations.

---

## [Decision Letter · Decision Letter 1]

18 May 2021

Associations between physical activity types and multi-domain cognitive decline in older adults from the Three-city cohort.

PONE-D-21-03103R1

Dear Dr. Carriere,

We’re pleased to inform you that your manuscript has been judged scientifically suitable for publication and will be formally accepted for publication once it meets all outstanding technical requirements.

Kind regards,

Masaki Mogi

Academic Editor

PLOS ONE

Additional Editor Comments (optional):

No further comment.

Reviewers' comments:

Reviewer's Responses to Questions

**Comments to the Author**

1. If the authors have adequately addressed your comments raised in a previous round of review and you feel that this manuscript is now acceptable for publication, you may indicate that here to bypass the “Comments to the Author” section, enter your conflict of interest statement in the “Confidential to Editor” section, and submit your "Accept" recommendation.

Reviewer #1: All comments have been addressed

Reviewer #2: (No Response)

2. Is the manuscript technically sound, and do the data support the conclusions?

Reviewer #1: Yes

Reviewer #2: (No Response)

3. Has the statistical analysis been performed appropriately and rigorously? 

Reviewer #1: Yes

Reviewer #2: (No Response)

4. Have the authors made all data underlying the findings in their manuscript fully available?

Reviewer #1: Yes

Reviewer #2: (No Response)

5. Is the manuscript presented in an intelligible fashion and written in standard English?

Reviewer #1: Yes

Reviewer #2: (No Response)

6. Review Comments to the Author

Reviewer #1: (No Response)

Reviewer #2: (No Response)

7. PLOS authors have the option to publish the peer review history of their article (what does this mean?). If published, this will include your full peer review and any attached files.

Reviewer #1: No

Reviewer #2: No

---

## [Editor Report · Acceptance letter]

21 May 2021

PONE-D-21-03103R1 

Associations between physical activity types and multi-domain cognitive decline in older adults from the Three-city cohort. 

Dear Dr. Carrière:

I'm pleased to inform you that your manuscript has been deemed suitable for publication in PLOS ONE. Congratulations! Your manuscript is now with our production department. 

Kind regards, 

on behalf of

Dr. Masaki Mogi 

Academic Editor

PLOS ONE